# CORRECT-BY-DESIGN SAFETY CRITICS USING NON-CONTRACTIVE BINARY BELLMAN OPERATORS

## ABSTRACT

The inability to naturally enforce safety in Reinforcement Learning (RL), with limited failures, is a core challenge impeding its use in real-world applications. One notion of safety of vast practical relevance is the ability to avoid (unsafe) regions of the state space. Though such a safety goal can be captured by means of an action-value-like function, a.k.a. safety critics, the associated operator lacks the desired contraction and uniqueness properties that the classical Bellman operator enjoys. In this work, we overcome the non-contractiveness of safety critic operators by leveraging the fact that safety is a binary property. To that end, we study the properties of the binary safety critic associated with a deterministic dynamical system that seeks to avoid reaching an unsafe region. We formulate the corresponding binary Bellman equation (B2E) for safety and study its properties. While the resulting operator is still non-contractive, we provide a full characterization of the fixed points representing–except for a spurious solution–maximal persistently safe regions of the state space that can always avoid failure. Interestingly, while maximality is often a desired notion for performance, in the context of safety, it means that the learned classification boundary is dangerously close and often crosses the region where failure is unavoidable. We thus further propose a one-sided version of the B2E that allows for more robust fixed points that are non-maximal. Finally, we provide an algorithm that, by design, leverages axiomatic knowledge of safe data points to avoid spurious fixed points. We provide initial empirical validation of our theory, showing how the proposed safety critic outperforms existing solutions, particularly regarding the number of samples (and failures) needed to secure safe policies.

## 1 INTRODUCTION

The last decade has witnessed a resurgence of Reinforcement Learning (RL) as a core enabler of Artificial Intelligence (AI). Today, RL algorithms can provide astonishing demonstrations of super-human performance in multiple settings, such as Atari (Mnih et al., 2015), Go (Silver et al., 2016), StarCraft II (Vinyals et al., 2017), and even poker (Nichols et al., 2019). However, this super-human success in RL is overwhelmingly limited to *virtual domains* (particularly games), where not only one has a vast amount of data and computational power, but also there is little consequence to failure in achieving a task. Unfortunately, physical domain applications (autonomous driving, robotics, personalized medicine) lack most of these qualities and are particularly sensitive to scenarios where the consequences of poor decision-making are catastrophic Yu et al. (2021),Brunke et al. (2022).

Guaranteeing safety in an RL setting is a challenging task, as agents often lack a priori knowledge of the safety of states and actions (Gu et al., 2022). Inspired by these challenges, numerous methods have been proposed to imbue RL methods with safety constraints, including expectation constraints (Paternain et al., 2022; Castellano et al., 2023), probabilistic/conditional value at risk constraints (Chow et al., 2017; Chen et al., 2023), and stability constraints (Li & Bastani, 2020; Taylor et al., 2020), among others. Such methods naturally lead to different safety guarantees, some of which can be theoretically characterized (Robey et al., 2020; Castellano et al., 2022). However, the majority of these methods fail to capture the safety-critical nature of some types of events that must be avoided at all costs, i.e., with probability one.

One type of safety constraint of practical relevance in safety-critical applications is reachability constraints (e.g.Bertsekas (1972); Sontag (2013), Ch. 3; Bansal et al. (2017)), wherein one seeks to avoid regions of the state space that are associated with failure events by computing sets that are either, persistently safe (a.k.a. control invariant safe sets (Gurriet et al., 2018)), i.e., regions of

the state that can avoid failure regions *for all times* by proper choice of actions, or unsafe regions (a.k.a. as backward reachable tubes (Mitchell, 2007)) where *failure is unavoidable* irrespectively of the actions taken. Recent research efforts incorporating such constraints in RL problems have shaped the notion of safety critics (Fisac et al., 2019; Srinivasan et al., 2020; Thananjeyan et al., 2021), which aim to compute action value-like functions that, based on information about either the (signed) distance to failure or a logical fail/not fail feedback, predict whether certain state-action pair is safe to take or is likely to lead to catastrophic failures.

Unfortunately, the computation and learning of safety critics is a challenging task since their corresponding Bellman-like equations (and associated operators) lack typical uniqueness (resp. contraction) properties that guarantee the validity of the solution (and convergence of RL algorithms). As a result, most works seek to compute approximate safety critics by introducing an artificial discount factor (Fisac et al., 2019; Hsu et al., 2021). This approximation, however, can have drastic effects on the accuracy of the critic, as approximately safe sets are not, by design, safe.

**Contributions of our work**   In this work, we seek to overcome the difficulties in computing accurate safety critics by developing supporting theory and algorithms that allow us to learn accurate and more robust safety critics directly from the original non-contractive safety critic operator. Precisely, we consider a setting with deterministic, continuous state dynamics that are driven by stochastic policies on discrete action spaces, and model safety as a binary (`safe`/`unsafe`) quantity. Building on the literature of risk-based safety critics, we develop a deeper theoretical understanding of the properties of the corresponding *binary safety (action-)value function* and how to exploit them to learn accurate safety critics. In doing so, we make the following contributions.

- **Characterization of solutions to the binary Bellman equations for safety** We study the properties of the *action-value function* associated with the binary safe/unsafe feedback and formulate a *binary Bellman equation* (B2E) that such function must satisfy. This binary Bellman equation is undiscounted and has a non-contractive operator with multiple fixed points. Nevertheless, we show (Theorem 1) that all (but one) of the possibly infinite solutions to the B2E represent regions of the state space that are: *(i)* persistently safe regions that can avoid failure for all future times and *(ii)* maximal, in the sense that no state that is declared to be unsafe can reach the declared safe region.

- **One-sided binary Bellman equations to compute non-maximal, persistently safe regions** While non-spurious solutions to the B2E represent valid, persistently safe regions, the maximality property ensures that the classification boundary often lies exactly at the edge of the unsafe region, making such a solution non-robust. This motivates the introduction of a one-sided B2E (O-B2E) that only requires solutions to satisfy the persistently safe property (and not maximality) (Theorem 2). The novel O-B2E induces a *set-valued* operator that drastically increases the number of fixed points and allows for solutions whose classification boundary has a larger margin from the truly unsafe boundary.

- **Algorithm for learning fixed points of a non-contractive set-valued operator** Finally, we provide an algorithm that is able to find a fixed point of the novel set-valued operator despite the lack of contraction. Our algorithm has two distinctive features that make this possible. First, it uses *axiomatic data points*, i.e., points of the state space that are a priori known to be safe. Secondly, it uses a classification loss that enforces *self-consistency* of the one-sided Bellman equation across samples. Preliminary numerical evaluations indicate that our proposed methodology outperforms a well-known safety critic (Fisac et al., 2019) in a simple setup, and show good performance in a 32-dimensional environment.

## 2   PROBLEM FORMULATION

**Environment**   We consider a Markov Decision Process $\langle \mathcal{S}, \mathcal{A}, F, \mathcal{G}, i, \rho \rangle$ where the state space $\mathcal{S}$ is *continuous* and compact, the action space $\mathcal{A}$ is *discrete* and finite, the map $F : \mathcal{S} \times \mathcal{A} \to \mathcal{S}$ is a *deterministic* transition function. The set $\mathcal{G}$ represents a set of "failure" states to be avoided. At each time step the agent receives as feedback the *insecurity* of state $s_t$, that is $i(s_t) = \mathbb{1}\{s_t \in \mathcal{G}\} \in \{0, 1\}$. Episodes start at a state $s_0 \sim \rho$ and run indefinitely or end when the system enters $\mathcal{G}$.

**Policies**   We consider stochastic, stationary policies $\pi : \mathcal{S} \to \Delta_{\mathcal{A}}$, and denote $\pi(a|s)$ the probability of picking $a \in \mathcal{A}$ when at state $s \in \mathcal{S}$. Since $\mathcal{A}$ is discrete and finite, and the transition dynamics are deterministic, the set of reachable states starting from any $s$ is finite as well, as defined next.

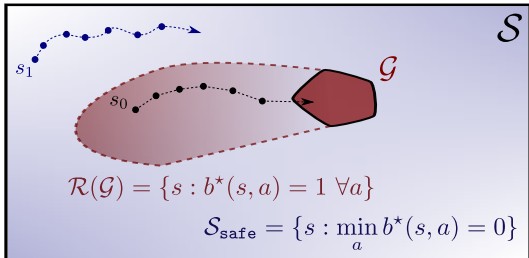

Figure 1: The optimal $b^\star$ describes different regions of the state space. The set $\mathcal{G}$ (solid, dark red) is the one to be avoided at all times. Due to the system dynamics, there is a region of the state space $\mathcal{R}(\mathcal{G})$ (shaded red) such that any trajectory starting there (e.g., from $s_0$) will inevitably enter $\mathcal{G}$. For any point in its complement $\mathcal{S}_{\texttt{safe}}$ (e.g. $s_1$), the optimal policy avoids $\mathcal{G}$ at all times.

**Definition 1 ($t$-step reachable sets)** *For any policy $\pi$ and any state $s \in \mathcal{S}$, the $t$-step reachable set from $s$ under $\pi$ is $\mathcal{F}_t^\pi(s) \triangleq \big\{ s' \in \mathcal{S} : \mathbb{P}^\pi\left(s_t = s' \mid s_0 = s\right) > 0 \big\}$. Similarly, for any $a \in \mathcal{A}$ we define $\mathcal{F}_t^\pi(s,a) \triangleq \big\{ s' \in \mathcal{S} : \mathbb{P}^\pi\left(s_t = s' \mid s_0 = s, a_0 = a\right) > 0 \big\}$.*

Given these notions of reachable sets, we can define the binary safety value functions for any policy.

**Definition 2 (Binary safety value functions)** *The binary safety (action-)value function of policy $\pi$ at state $s$ (and action $a$) is:*

$$v^\pi(s) \triangleq \sup_{t \geq 0} \max_{s_t \in \mathcal{F}_t^\pi(s)} i(s_t), \quad \left( b^\pi(s,a) \triangleq \sup_{t \geq 0} \max_{s_t \in \mathcal{F}_t^\pi(s,a)} i(s_t) \right) \tag{1}$$

We choose the notation $b(\cdot, \cdot)$ instead of the usual $Q$ to emphasize that it is a binary action-value function. Note that $b^\pi(s,a) = 1$ if and only if starting from $(s,a)$ and following $\pi$, there is positive probability of entering $\mathcal{G}$. The optimal (action-)value functions are then defined.

**Definition 3 (Optimal binary value functions)** *For all $s \in \mathcal{S}$ and $a \in \mathcal{A}$, the optimal value and action-value functions are $v^\star(s) \triangleq \min_\pi v^\pi(s)$ and $b^\star(s,a) \triangleq \min_\pi b^\pi(s,a)$.*

**Relationship between safety and the optimal binary functions**  These optimal value functions fully characterize the logical `safe`/`unsafe` nature of each state or state-action pair, and have nice interpretations in terms of how they partition the state-space, as illustrated in Fig. 1. Recall that the safety goal is to avoid $\mathcal{G}$. However, due to the MDP dynamics, this might not be possible for every state outside $\mathcal{G}$[1]. A state $s$ is persistently safe if trajectories from $s$ can avoid $\mathcal{G}$ at all times—in other words, if $\exists a \in \mathcal{A} : b^\star(s,a) = 0$. Conversely, a state $s$ is doomed to fail if $b^\star(s,a) = 1 \, \forall a \in \mathcal{A}$. We use $\mathcal{R}(\mathcal{G})$ to denote this set of "unsafe states" that are doomed to fail. The complement of this set is the set of persistently safe states, and the "safe" actions for each state are given by:

$$\mathcal{S}_{\texttt{safe}} = \Big\{ s \in \mathcal{S} : \min_{a \in \mathcal{A}} b^\star(s,a) = 0 \Big\} \qquad \mathcal{A}_{\texttt{safe}}(s) = \big\{ a \in \mathcal{A} : b^\star(s,a) = 0 \big\}. \tag{2}$$

Just like in the standard RL setup, each (action-)value function has associated Bellman equations.

**Proposition 1 (Binary Bellman Equations)** *For any policy $\pi$, the following set of Bellman equations hold for all $s \in \mathcal{S}$, for all $a \in \mathcal{A}$: $b^\pi(s,a) = i(s) + \big(1 - i(s)\big)v^\pi(s')$, where $b^\pi(s,a) = i(s) + \big(1 - i(s)\big)v^\pi(s')$. In particular, any optimal policy satisfies:*

$$b^\star(s,a) = i(s) + \big(1 - i(s)\big) \min_{a' \in \mathcal{A}} b^\star(s',a'). \tag{3}$$

*Proof: See Appendix A.2.* □

**Unsafety as a logical OR**  The Bellman equation for the optimal $b^\star$ can be understood as: "an $(s,a)$ pair is *unsafe* ($b^\star(s,a) = 1$) if either: the current state is unsafe ($i(s) = 1$), OR it leads to an unsafe state later in the future ($\min_{a'} b^\star(s',a') = 1$)."

---

[1] A car heading to a wall ($\mathcal{G}$) one meter away at 100mph will hit it, regardless of the actions taken.

**Non-contractive Bellman operator**    The optimal binary function of equation 3 has an associated operator, acting on the space of functions $\mathcal{B} = \{b : \mathcal{S} \times \mathcal{A} \to \{0, 1\}\}$:

$$\mathcal{T} : \mathcal{B} \to \mathcal{B} : (\mathcal{T}b)(s,a) = i(s) + (1 - i(s)) \min_{a' \in \mathcal{A}} b(s', a') \quad \forall(s,a) \in \mathcal{S} \times \mathcal{A} \qquad (4)$$

One of the key features in the standard (discounted) Bellman equations for infinite-horizon problems is that it has an associated operator that is contractive (Bertsekas, 2015, p.45), and as such, it admits a unique fixed point (the optimal value function). This is crucial for the application of value iteration procedures or for methods reliant on temporal differences (Schwartz, 1993). Surprisingly, the operator defined in equation 4 is non-contractive, and as such, it admits more fixed points than the optimal $b^\star$. In particular, there are even fixed points of equation 4 that have no physical meaning. We will soon see, however, that all–except for one–of them do have a physical interpretation.

## 2.1 CLOSELY RELATED WORK

**Control-theoretic approaches for computing $\mathcal{S}_{\texttt{safe}}$**    Standard tools from Control Theory exist to approximate the safe regions corresponding to $b^\star(\cdot, \cdot)$, both for linear (Girard et al., 2006) and non-linear dynamics (Mitchell & Templeton, 2005). The latter requires knowledge of the transition map $F(\cdot, \cdot)$ along with the signed distance to the unsafe region (Mitchell et al., 2005). This amounts to solving partial differential equations (PDEs) of the Hamilton-Jacobi-Isaacs (HJI) type (Bansal et al., 2017), and yields value functions whose zero super-level sets correspond to $\mathcal{S}_{\texttt{safe}}$.

**Risk-based vs Reachability-based safety critics**    The binary action-value function $b^\star$ defined in equation 3 is closely related to recent work on *Risk-based* safety critics (Srinivasan et al., 2020; Thananjeyan et al., 2021), which use binary information to indicate the risk of unsafe events. However, unlike risk-based critics, which seek to measure a cumulative expected risk $b^\star_{\text{risk}}(s,a) = \max_\pi E_\pi[\sum_{k=t}^\infty \gamma^{k-t} i(s_t) | s_t = s, a_t = a] \in [0,1]$, our binary critic only takes values $b^*(s,a) \in \{0, 1\}$, and outputting 1 whenever unavoidable failure has positive probability. *Reachability based* safety critics, build on the literature of HJI equations and seek to measure the largest (signed) distance $h(s_t)$ that one can sustain from the failure set $\mathcal{G}$, i.e., $b^\star_{\text{reach}}(s,a) = \sup_\pi \inf_{t \geq 0} h(s_t)$ almost surely (Fisac et al., 2019). Our binary critic $b^\star$ is indeed related to $b^\star_{\text{reach}}$, when the signed distance $h(s)$ is replaced with the binary signal $-i(s)$. We will soon show that this particular choice of safety measure allows for a precise characterization of the fixed points of equation 4.

**To contract or not to contract**    Despite the diversity of safety critics present in the literature, a common practice in both risk-based critics (Srinivasan et al., 2020; Thananjeyan et al., 2021) and reachability-based critics (Fisac et al., 2019; Chen et al., 2021) is the introduction of a discount factor $\gamma < 1$. While this leads to desired uniqueness and contraction properties for the operator, it comes with trade-offs: it degrades the accuracy, requiring the introduction of conservative thresholds Srinivasan et al. (2020); Chen et al. (2021), which further limits exploration. Notably, such an approach is particularly worrisome when seeking to guarantee persistent safety (the ability to avoid failure for all future times), as such property is not preserved for finite accuracy approximation, even for thresholded ones. In this work, we overcome this limitation by seeking to learn directly using the non-contractive operator, thus guaranteeing, by design, the correctness of the solution.

## 3 BINARY CHARACTERIZATION OF SAFETY

The fixed points $b^\star$ of the binary Bellman operator have a meaningful interpretation in terms of the topology of the state-space, and can be used to derive persistently safe policies. This connection will be better understood once we define the notion of control invariant safe sets.

**Definition 4 (Control invariant safe (CIS) set)** *A set $\mathcal{C} \subset \mathcal{S}$ is a control invariant safe (CIS) set if there exists a policy $\pi$ such that:*

*i) (Control invariance):* $\forall s_0 \in \mathcal{C}, \forall t \geq 0, \quad \mathcal{F}_t^\pi(s_0) \subset \mathcal{C}$

*ii) (Safety):* $\qquad\qquad\quad \forall s_0 \in \mathcal{C}, \forall t \geq 0, \quad \mathcal{F}_t^\pi(s_0) \cap \mathcal{G} = \emptyset.$

In essence, *(i)* means that there exists a controller that guarantees that trajectories starting in $\mathcal{C}$ can be made to remain in $\mathcal{C}$ forever, which is a standard notion in control theory (Bertsekas, 1972; Blanchini, 1999). Property *(ii)* means this can be done while also avoiding the unsafe region $\mathcal{G}$!

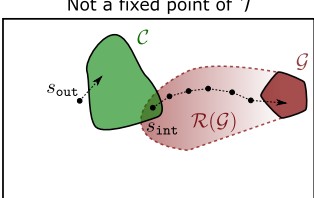
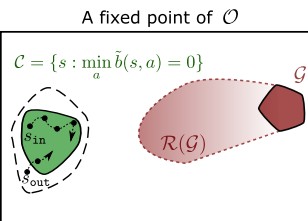

Figure 2: An illustration of Thm. 1 and Thm. 2 Left: a valid fixed point $\tilde{b}$ of $\mathcal{T}$ and its corresponding safe control invariant set. Trajectories starting in $\mathcal{C}$ can be made to remain in $\mathcal{C}$. Middle: a function $\tilde{b}$ that is not a fixed point. A state $s_{\text{int}}$ in the intersection will inevitably lead to the unsafe region $\mathcal{G}$, so $\tilde{b}(s, a)$ should be 1 for all states in the trajectory (which would mean $s_{\text{int}} \notin \mathcal{C}$). Similarly, a state $s_{\text{out}}$ outside $\mathcal{C}$ cannot reach inside. If it could, $\tilde{b}(s_{\text{out}}, a) = 1$ for some $a \in \mathcal{A}$, but it would transition to a state where $\min_{a'} \tilde{b}(s', a') = 0$, violating equation 3. Right: a valid fixed point of the one-sided operator $\mathcal{O}$ of Thm. 2. States starting in $\mathcal{C}$ can be made to remain there; there is no guarantee that a state in $\mathcal{C}^{\complement}$ cannot enter $\mathcal{C}$. This set is CIS, and a subset from the fixed point of $\mathcal{T}$.

With these definitions in place, we are ready for the main result of this section.

**Theorem 1 (Fixed points and control invariant safe sets)** *Let $\tilde{b} : \mathcal{S} \times \mathcal{A} \to \{0, 1\}$ be a fixed point of equation 4. Then either $\tilde{b}(s, a) = 1 \ \forall (s, a)$ (spurious fixed point), or:*

   *i) $\mathcal{C} \triangleq \{ s \in \mathcal{S} : \min_a \tilde{b}(s, a) = 0 \}$ is control invariant safe (CIS).*

   *ii) $\mathcal{C}$ is unreachable from outside, i.e., $\mathcal{F}_t^\pi (s_0) \cap \mathcal{C} = \emptyset \qquad \forall s_0 \in \mathcal{S} \setminus \mathcal{C}, \forall \pi, \forall t \geq 0$.*

   *iii) Any policy $\pi$ that satisfies equation 5 renders $\mathcal{C}$ CIS.*

$$\tilde{b}(s, a) = 1 \ \Rightarrow \pi(a|s) = 0, \ \forall s \in \mathcal{C}. \tag{5}$$

*Proof: The proof is in Appendix A.3.* $\qquad\qquad\qquad\qquad\qquad\qquad\qquad\qquad\qquad\qquad$ □

The first statement proclaims that, starting in $\mathcal{C}$, the system can be made to remain in $\mathcal{C}$ forever (thus ensuring safety). The contrapositive of property *(ii)* sheds light on a notion of *maximality* of $\mathcal{C}$:

**Corollary 1 (Maximality of the CIS set)** *Let $\mathcal{X}$ be a strict subset of $\mathcal{C}$. If $\mathcal{X}$ is reachable[2] from $\mathcal{C} \setminus \mathcal{X}$, then $\mathcal{X}$ cannot be associated[3] with any fixed point of equation 4.*

We refer the reader to Fig. 2 for an illustration of valid and nonvalid fixed points. By means of Theorem 1 and Corollary 1 we achieve our goal of identifying the fixed points of the binary Bellman operator to maximal persistently safe states. In the next section, we relax the binary Bellman operator to increase the number of fixed points and safe regions associated with them, making it simpler to find safe policies.

## 4   SAFETY THROUGH A ONE-SIDED OPERATOR

Theorem 1 states that any non-spurious fixed point of equation 4 yields a CIS set, along with a policy that guarantees said invariance. This set is maximal (in the sense of Corollary 1), and cannot be reached from outside. While maximality is a desired property, trying to learn the boundary of such maximal CIS sets $\mathcal{C}$ under limited data and with high fidelity is challenging. Moreover, overestimating $\mathcal{C}$ with an approximate set $\hat{\mathcal{C}}$ would make the unsafe region $U = \hat{\mathcal{C}} \setminus \mathcal{C}$ to be declared safe. To prevent this problem, we will avoid the boundary, aiming for inner safe sets included in $\mathcal{C}$. Thus, we will sacrifice the maximality given by property *(ii)* in Theorem 1, and focus on the safety property *(i)*.

---

[2]i.e. if $\exists \pi, \exists t \geq 0, \exists s_0 \in \mathcal{C} \setminus \mathcal{X} : \mathcal{F}_t^\pi (s_0) \cap \mathcal{X} \neq \emptyset$

[3]that is to say: $\forall \tilde{b} : \tilde{b} = \mathcal{T}\tilde{b}, \mathcal{X} \neq \{ s \in \mathcal{S} : \min_a \tilde{b}(s, a) = 0 \}$

In this direction, we relax the binary Bellman equations equation 3 to yield fixed points that only certify property *(i)*. As such, for any $(s, a) \in \mathcal{S} \times \mathcal{A}$ and $s' = F(s, a)$, we want a function satisfying:

$$b(s, a) \geq i(s) + (1 - i(s)) \min_{a' \in \mathcal{A}} b(s', a'), \tag{6}$$

This inequality has an associated *set-valued* operator mapping functions into sets of functions.

**Definition 5 (One-sided operator)** *Let $\mathcal{N}(\mathcal{B})$ denote the class of non-empty subsets of $\mathcal{B} = \{b : \mathcal{S} \times \mathcal{A} \to \{0, 1\}\}$ . We define the set-valued, one-sided operator $\mathcal{O} : \mathcal{B} \to \mathcal{N}(\mathcal{B})$ as:*

$$(\mathcal{O}b) = \left\{ b' \in \mathcal{B} : b'(s, a) - i(s) - (1 - i(s)) \min_{a' \in \mathcal{A}} b(s', a') \geq 0 \quad \forall (s, a) \in \mathcal{S} \times \mathcal{A} \right\} \tag{7}$$

*A binary function $\tilde{b}$ is a fixed point of equation 7 iff $\tilde{b} \in (\mathcal{O}\tilde{b})$.*

Given a fixed point $\tilde{b}$ of the one-sided operator $\mathcal{O}$, the pair $(s, a)$ could be declared unsafe by $\tilde{b}$ even if $s$ is safe ($i(s) = 0$) and the next state $s'$ can be driven to safety as well, i.e., $\tilde{b}$ can be potentially conservative in describing the persistently safe region. As the next theorem shows, the fixed points of this operator have, indeed, the desired CIS property. Moreover, as shown in Fig. 2 and stated next, the set may no longer be maximal as it could potentially be reached from outside.

**Theorem 2 (Fixed points of the one-sided operator)** *Let $\tilde{b} : \mathcal{S} \times \mathcal{A} \to \{0, 1\}$ be a fixed point of equation 7. Then either $\tilde{b}(s, a) = 1 \; \forall (s, a)$ (spurious fixed point), or:*

  *i)* $\mathcal{C} \triangleq \left\{ s \in \mathcal{S} : \min_a \tilde{b}(s, a) = 0 \right\}$ *is control invariant safe (CIS).*

  *ii) Any policy $\pi$ that satisfies equation 5 renders $\mathcal{C}$ CIS.*

*Proof: The proof is in Appendix A.4* $\qquad\qquad\qquad\qquad\qquad\qquad\qquad\qquad\qquad\qquad\qquad$ $\square$

Theorem 2 proves that the fixed points of equation 7 and their associated sets retain the desired CIS property. In addition, the one-sided operator can accommodate more fixed points, allowing for inner approximations of the maximal sets whose classification boundaries have a larger margin from the truly unsafe boundary. In the following section, we leverage these results using the one-sided operator to build an algorithm that aims to find fixed points of $\tilde{\mathcal{O}}$.

---

**Algorithm 1:** Pseudocode for learning the binary value function

---

**Input:** Safe dataset $\mathcal{D}_{\texttt{safe}}$;
**Output:** Binary value function $b^\theta(\cdot, \cdot)$;

1 Initialize $b^\theta(\cdot, \cdot)$ using $\mathcal{D}_{\texttt{safe}}$ and $\mathcal{M} = [\,]$ ;             ▷ `Transition buffer.`
2 **repeat**
3     **for** *i=0,…NUM_EPISODES-1* **do**
4        Run episodes, store $\left(s_k, a_k, i(s_k), s'_k\right)_{k=1}^K$ transitions in $\mathcal{M}$;
5     **end**
6     $\mathcal{D}_{\texttt{unsafe}} \leftarrow$ `build_unsafe_dataset`$(b^\theta, \mathcal{M})$; ▷ `Use` $b^\theta$ `to compute labels.`
7     Build $\mathcal{D} = \mathcal{D}_{\texttt{safe}} \cup \mathcal{D}_{\texttt{unsafe}}$ ;                    ▷ `Complete dataset.`
8     **repeat**
9        Run gradient steps on $\mathcal{L}_{\texttt{train}}$ ;                  ▷ `Update` $b^\theta$
10     **until** *Accuracy*$(b^\theta, \mathcal{D}) = 1$;
11     $\mathcal{D}_{\texttt{unsafe}} \leftarrow$ `build_unsafe_dataset`$(b_i, \mathcal{M})$;     ▷ $b^\theta$ `has changed w.r.t.`**6**
12     Build $\mathcal{D} = \mathcal{D}_{\texttt{safe}} \cup \mathcal{D}_{\texttt{unsafe}}$ ;                  ▷ `New dataset`
13     **if** *Accuracy*$(b^\theta, \mathcal{D}) \neq 1$;            ▷ `Check consistency of B2E`
14     **then**
15        **go to** Line **8** ;     ▷ `Not self-consistent` $\Rightarrow$ `Re-train the network`
16     **end**
17 **until** *termination*;

---

### 4.1 ALGORITHM

We propose learning fixed points of $\mathcal{O}$ in equation 7 by training a neural network classifier. We will denote the learned function by $b^\theta(\cdot, \cdot)$ where $\theta$ collects the parameters of the network. The network takes as input each state and outputs the value $b^\theta(s, a)$ for each of the possible actions. The last layer is a point-wise sigmoid activation function ensuring $b^\theta(s, a)$ lies in the unit interval. We use $\hat{b}^\theta(s, a) \triangleq \texttt{Round}\left(b^\theta(s, a)\right)$ to denote the predicted label. Note that our threshold (at 1/2) will be fixed during training and testing. The pseudocode for the main algorithm can be found in Alg. 1. We provide a comprehensive breakdown of its main components next.

**Dataset:** The dataset $\mathcal{D}$ consists of $(s, a, y)$ tuples, where $y$ is a $\{0, 1\}$ label, and has two components. A prescribed [4] safe set $\mathcal{D}_{\texttt{safe}}$ (for which $y = 0$) and a dynamically updated $\mathcal{D}_{\texttt{unsafe}}$ of unsafe transitions detected during data collection. We have observed empirically that the addition of $\mathcal{D}_{\texttt{safe}}$ helps prevent the collapse to the trivial fixed point described in Theorem 1. The algorithm iterates over the following three loops:

**Environment interaction:** Episodes start from a state $s_0$ sampled from the initial distribution $\rho$. To collect $(s, a, s', i(s))$ transitions and store them in a memory buffer $\mathcal{M}$ we run episodes by following a policy that satisfies equation 5, for example the *uniform safe policy*, which takes actions uniformly over the presumed-safe ones:

$$\pi^\theta(a|s) = \begin{cases} 0 & \text{if } \hat{b}^\theta(s, a) = 1 \\ 1/\sum_{a' \in \mathcal{A}} \mathbb{1}\{\hat{b}^\theta(s, a') = 0\} & \text{if } \hat{b}^\theta(s, a) = 0 \end{cases} \tag{8}$$

**Building the dataset:** After collecting transitions, the binary value function is used to compute labels via the right hand side of equation 3, that is, $y_k^\theta = i(s) + (1 - i(s)) \min_{a'} b^\theta(s_k', a')$ for all $(s_k, a_k, i(s_k), s_k') \in \mathcal{M}$. Note that these are "soft" labels $y_k^\theta \in [0, 1]$. Those that satisfy $y_k^\theta \geq \frac{1}{2}$ are added to $\mathcal{D}_{\texttt{unsafe}}$. This procedure is dubbed $\texttt{build\_unsafe\_dataset}(b, \mathcal{M})$ in Algorithm 1.

**Training the network:** The network is trained by running mini-batch gradient descent on the binary cross-entropy loss until it can correctly predict all the labels in $\mathcal{D} := \mathcal{D}_{\texttt{safe}} \cup \mathcal{D}_{\texttt{unsafe}}$. Once that is achieved, the labels in $\mathcal{D}_{\texttt{unsafe}}$ are re-computed (some might have changed since $b^\theta$ was updated during this process), and the program checks whether it can correctly predict the labels again. It repeats this process until all labels are predicted correctly, yielding a binary function that is self-consistent across the whole dataset.

## 5 NUMERICAL EXPERIMENTS

We present numerical validations of our algorithm on two different environments. We contrast our method first against SBE (Fisac et al., 2019), a well-known safety-critic, and against PPO (Schulman et al., 2017), a state-of-the-art RL algorithm.

### 5.1 INVERTED PENDULUM

We begin by showcasing our algorithm on a modified version of the inverted pendulum problem (Towers et al., 2023). We choose this environment because it allows easy visualization of the learned control invariant safe sets, and because these can be compared against numerically obtained "grounds truth" references.

**Environment** The state of the system $s = [\theta, \ \omega]^\top$ is the angular position and angular velocity of the pendulum with respect to the vertical. The action $a \in [-a_{\max}, a_{\max}]$ is the torque applied on the axis, which we discretize in 5 equally spaced values. The **goal** in this task is to avoid falling past the horizontal, i.e. $\mathcal{G} = \{(\theta, \omega) : |\theta| \geq \frac{\pi}{2}\}$.

---

[4]e.g. $(s, a)$ pairs close to the equilibrium of the system, or sampled trajectories from a known, safe policy.

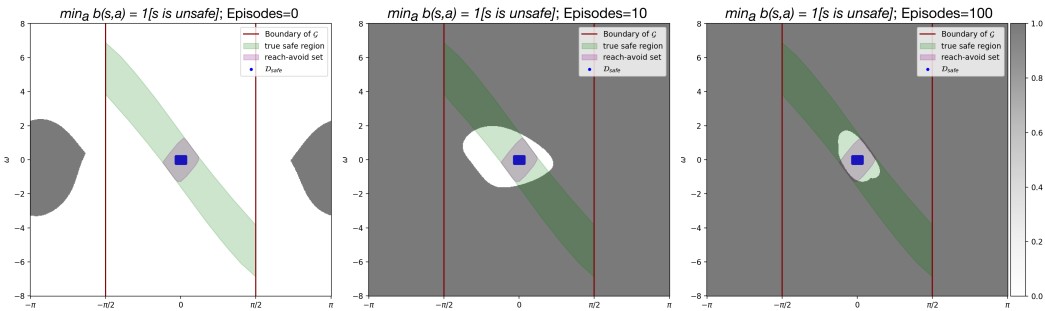

Figure 3: Learned safe regions for the inverted pendulum problem during early (left, middle) and latter (right) stages of training. The white area corresponds to states classified as safe. The solid maroon lines show the boundary of the unsafe region $\mathcal{G}$ (falling past the horizontal). The green region shows the set of states that can avoid $\mathcal{G}$ at all times, and the purple region shows the set of safe states reachable from $\mathcal{D}_{\texttt{safe}}$. These two sets were computed numerically using an optimal control toolbox (Mitchell & Templeton, 2005). As learning progresses, the classifier learns a control invariant safe set inside the green region. Animations at `https://tinyurl.com/6u8fvaux`.

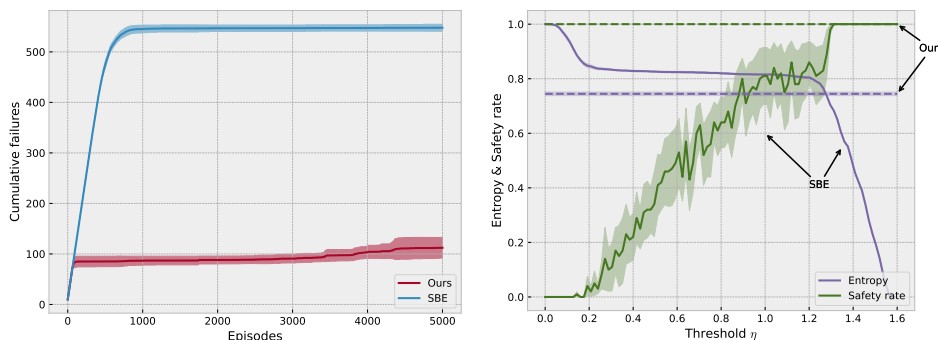

Figure 4: Left: cumulative failures during training of our algorithm (red) and SBE (blue) for the inverted pendulum. Solid lines represent the means across 5 seeds, shaded areas are $95\%$ confidence intervals. Our algorithm is 5 times safer. Right: safety rate (fraction of safe episodes) and *entropy* of each learned model. Our algorithm (shaded lines) always uses the uniform safe policy. SBE is tested for different threshold values $\eta$. Our policy is $100\%$ safe and is exploratory (high entropy). Only the most conservative SBE policies (large $\eta$) are $100\%$ safe, but have low entropy (limited exploration).

**Training protocol** We take $\mathcal{D}_{\texttt{safe}}$ to be a small grid of $(s, a)$-pairs near the unstable equilibrium. Episodes are started from $\mathcal{D}_{\texttt{safe}}$ and end whenever the pendulum reaches the unsafe region, or after 200 steps. The behavioral policy is the "uniform safe" as defined in equation 8. We alternate between collecting data for 10 episodes, building the dataset and training the network as explained in Sec. 4.1. Details on network architecture and hyperparameters are relegated to the Appendix A.5.

**Ground truth** We compare the safe region learned by our algorithm against ground truths computed numerically with optimal control tools (Mitchell & Templeton, 2005). Fig. 3 shows in green (resp. light gray) the maximum CIS set in the entire state (resp. the maximum CIS for trajectories that start in the support of $\rho$). The learned safe region (in white) at different stages of training is also shown. At the beginning, the network is only fit to $\mathcal{D}_{\texttt{safe}}$. As episodes run and it collects unsafe transitions, it effectively learns a CIS set included in the true safe region for the problem.

**Training performance** We benchmark our proposed methodology against the Safety Bellman Equation (SBE) of Fisac et al. (2019). This algorithm learns a safety-critic $q(s, a)$ and considers "safe" those actions that have $q(s, a) \geq \eta$, for a threshold $\eta$. Hyperparameters for that algorithm are taken from Hsu et al. (2021) and can be seen in Appendix A.5. Fig. 4 (left) shows the cumulative failures during training; (a *failure* is an episode that touched the unsafe region $\mathcal{G}$). Our algorithm is clearly safer during training.

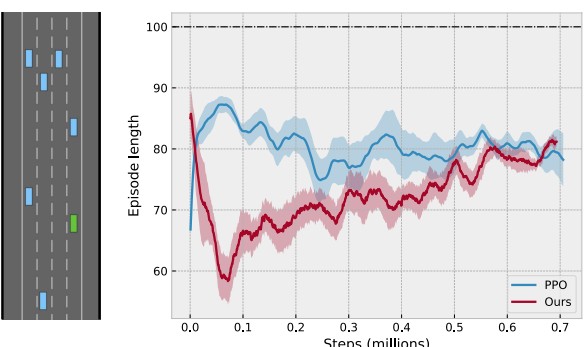

Figure 5: Left: autonomous driving environment (Leurent, 2018). Right: Average episode length during training, for our algorithm and PPO. Results are averaged over 5 seeds. Solid line represents the mean, shaded area corresponds to $\pm$ the standard deviation.

**Post-training evaluation**   We evaluate the performance of each model after training and show it in Fig. 4 (right). We test the uniform safe policy of our model against the safety critic for SBE. In the latter, we consider—for varying threshold $\eta$—the safe policy that maximizes exploration, i.e., the uniform policy taking actions $a$ such that $q(s, a) \geq \eta$. We illustrate the *safety rate*, defined as the proportion of safe episodes, and the *average entropy* of each policy $\tilde{\mathcal{H}}_\pi \triangleq \mathbb{E}_{s \sim \mathcal{RA}} \left[ \mathcal{H} \left( \pi (\cdot \mid s) \right) \right]$, where $\mathcal{RA}$ is the set of safe states reachable from the origin (see 'reach-avoid' set in Fig. 3). Our algorithm obtains perfect safety rate, while SBE only achieves it for safer policies (large enough $\eta$). These latter policies, though safe, are less exploratory—i.e. smaller entropy—than ours. In summary, our achievements are twofold: we learn a *persistently safe* family of policies that is *more exploratory* than the SBE counterpart. As argued in Section 2.1, for traditional safety critics, there is no straightforward connection between the threshold $\eta$ and discount factor $\gamma < 1$ needed to achieve safe policies, and safety comes at the expense of less exploration, which is undesired and difficult to balance. The solution found with our algorithm strikes a good balance between safety and the richness of the class of policies guaranteed to be safe.

## 5.2   AUTONOMOUS DRIVING

We finish the experiment section by showing the applicability of our method in a high-dimensional, autonomous driving environment (Leurent, 2018), comparing against PPO (Schulman et al., 2017).

**Environment**   The observation space is 25 dimensional, corresponding to the position and relative velocities of vehicles on the highway. The goal is to drive the car while avoiding crashes with other vehicles (see Fig. 5, left). Further details of the environment in A.5.

**Performance comparison**   We set the horizon of this environment to 100, more than doubling its default value (Leurent, 2018). In this context, a safer policy is one that runs for longer without crashing. Fig. 5 on the right shows the episode length as a function of environment steps for our algorithm and PPO. Results are averaged over five runs. After 700.000 steps, our algorithm slightly outperforms PPO in terms of safety. This warrants special merit, since our algorithm learns a family of safe policies, while PPO only learns one.

## 6   CONCLUSION

In this work we proposed a framework for obtaining correct-by-design safety critics in RL, under the goal of always avoiding a region of the state space. Our framework exploits the logical safe/unsafe nature of the problem and yields binary Bellman equations with multiple fixed points. We argue that all these fixed points are meaningful, by characterizing their structure in terms of guaranteeing safety and maximality. We circumvent the challenge of obtaining a maximal one by introducing a one sided operator, whose solutions possess the desired safety properties. Numerical experiments vallidate our theory and show that we can safely learn safer, more exploratory policies.

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

# A APPENDIX

## A.1 SOME IDENTITIES

The $t$-step reachable set from $s$ is the union of the $(t-1)$-step reachable set from all the successor states of $s$:

$$\mathcal{F}_t^\pi(s) = \bigcup_{s' \in \mathcal{F}^\pi(s)} \mathcal{F}_{t-1}^\pi(s') \tag{9}$$

Furthermore, the $t$-step reachable set from $(s,a)$ is the $t-1$ set from the successor $s' = F(s,a)$:

$$\mathcal{F}_t^\pi(s,a) = \mathcal{F}_{t-1}^\pi(s') \qquad \forall t \geq 1 \tag{10}$$

## A.2 PROOF OF PROPOSITION 1

We will show:
$$b^\pi(s,a) = i(s) + \big(1 - i(s)\big)v^\pi(s') \qquad \text{where } s' = F(s,a).$$
The following identities hold, as explained below.

$$b^\pi(s,a) = \sup_{t \geq 0} \max_{s_t \in \mathcal{F}_t^\pi(s,a)} i(s_t) \tag{11}$$

$$= \max\left\{ i(s),\ \sup_{t \geq 1} \max_{s_t \in \mathcal{F}_t^\pi(s,a)} i(s_t) \right\} \tag{12}$$

$$= \max\left\{ i(s),\ \sup_{t \geq 1} \max_{s_t \in \mathcal{F}_{t-1}^\pi(s')} i(s_t) \right\} \tag{13}$$

$$= \max\left\{ i(s),\ \sup_{t \geq 0} \max_{s_t \in \mathcal{F}_t^\pi(s')} i(s_t) \right\} \tag{14}$$

$$= \max\{i(s), v^\pi(s')\} \tag{15}$$

The first identity is the definition of $b^\pi$. In equation 12 unroll the first step in $\sup\{\cdot\}$. Next use the identity of equation 10. Finally introduce the change of variables $t \leftarrow t-1$ and recognize $v^\pi(s')$.

Recall $b^\pi(s,a), i(s)$ and $v^\pi(s')$ are binary. We consider two cases.

If $i(s) = 1$:
$$i(s) = 1 \geq b^\pi(s,a) \geq i(s) \implies b^\pi(s,a) = i(s) \tag{16}$$

If $i(s) = 0$:
$$b^\pi(s,a) = \max\{0, v^\pi(s')\} = 1 \cdot v^\pi(s') = (1 - i(s))v^\pi(s') \tag{17}$$

Hence:
$$b^\pi(s,a) = i(s) + \big(1 - i(s)\big)v^\pi(s'),$$
which completes the first part of the proof.

Before proceeding to the last part of the proof, we note that a similar Binary Bellman equation holds for the value function (which we omitted in the manuscript for brevity):

**Proposition 2 (Binary Bellman equation for $v^\pi$)** *For all $\pi$, for all $s \in \mathcal{S}, a \in \mathcal{A}$:*

$$v^\pi(s) = i(s) + \big(1 - i(s)\big) \max_{s' \in \mathcal{F}^\pi(s)} v^\pi(s') \qquad \text{where } s' = F(s,a) . \tag{18}$$

*Proof: We omit the proof since it is virtually identical to equations 11–15.* $\square$

Now, going back to the proof of Proposition 1, remains to be shown:

$$b^\star(s,a) = i(s) + \big(1 - i(s)\big) \min_{a' \in \mathcal{A}} b^\star(s',a') .$$

In light of what we have just proved, it suffices to show the following Bellman optimality condition:

$$\min_{a \in \mathcal{A}} b^\star(s,a) = v^\star(s) . \tag{19}$$

We again consider two cases.

If $i(s) = 1$:

$$\forall \pi, \forall a \in \mathcal{A}, \quad 1 = v^\pi(s) = b^\pi(s, a) \implies v^\star(s) = \min_a b^\star(s, a),$$

so the result holds trivially.
If $i(s) = 0$:
Let $\pi^\star$ be an optimal policy. Then by Proposition 2:

$$v^\star(s) = \overbrace{i(s)}^{=0} + \overbrace{\big(1 - i(s)\big)}^{=1} \max_{s' \in \mathcal{F}^{\pi^\star}(s)} v^\star(s') \tag{20}$$

$$= \max_{s' \in \mathcal{F}^{\pi^\star}(s)} v^\star(s') \tag{21}$$

$$= \max_{a \in \mathrm{Supp}[\pi(\cdot|s)]} v^\star(F(s, a)) \tag{22}$$

$$\geq \min_{a \in \mathrm{Supp}[\pi(\cdot|s)]} v^\star(F(s, a)) \tag{23}$$

$$\geq \min_{a \in \mathcal{A}} v^\star(F(s, a)) \tag{24}$$

$$= \min_{a \in \mathcal{A}} \big[i(s) + \big(1 - i(s)\big) v^\star(F(s, a))\big] \tag{25}$$

$$= \min_{a \in \mathcal{A}} b^\star(s, a) \tag{26}$$

where the first inequality follows from $\max \geq \min$, and the second one for optimizing over a larger set.
Thus:

$$v^\star(s) \geq \min_{a \in \mathcal{A}} b^\star(s, a) \quad \forall s \in \mathcal{S}, \tag{27}$$

and we want to show that the result holds with equality. By contradiction, assume the inequality is strict for some $s \in \mathcal{S}$, that is to say:

$$\exists a^\dagger \in \mathcal{A} : v^\star(s) > b^\star(s, a^\dagger). \tag{28}$$

Since the inequality is strict, it must be that $v^\star(s) = 1$ and $b^\star(s, a^\dagger) = 0$.

Now consider a policy $\pi^\dagger$ similar to $\pi^\star$, but that only takes action $a^\dagger$ at state $s$:

$$\pi^\dagger : \begin{cases} \pi^\dagger(a_1|s) = 1 \\ \pi^\dagger(\cdot|s') = \pi^\star(\cdot|s') \quad \forall s' \neq s \end{cases} \tag{29}$$

We then have:

$$v^{\pi^\dagger}(s) = \max_{s' \in \mathcal{F}^{\pi^\dagger}(s)} v^{\pi^\dagger}(s') = v^{\pi^\dagger}\big(F(s, a^\dagger)\big) = b^\pi(s, a^\dagger) < v^\star(s), \tag{30}$$

hence $v^{\pi^\dagger}(s) < v^\star(s)$ which means $\pi^\star$ is not optimal. This is a contradiction. It must then be that equation 27 holds with equality, as was claimed.

### A.3 PROOF OF THM. 1

*Proof:* Throughout this proof, we will make use of the following alternative representation of fixed points of the binary Bellman operator $\mathcal{T}$.

**Lemma 1** $\tilde{b}$ *is a fixed point of* $\mathcal{T}$ *if and only if it satisfies, for all* $s \in \mathcal{S}$*, for all* $a \in \mathcal{A}$*:*

$$\tilde{b}(s, a) = \max\left\{i(s), \min_{a' \in \mathcal{A}} \tilde{b}(s', a')\right\} \quad \text{where } s' = F(s, a). \tag{31}$$

*Proof: The proof follows from equations 11–15 in Proposition 1 applied to* $\pi^\star$. $\qquad\square$

Now, to the main proof.

**Spourious fixed point** Firstly, note that $\tilde{b} \equiv 1$ is indeed one possible fixed point of equation 3:

$$\forall (s,a) \in \mathcal{S} \times \mathcal{A}, \quad 1 = \tilde{b}(s,a) = i(s) + \big(1 - i(s)\big) \min_{a'} \tilde{b}(s',a') \geq \min_{a'} \tilde{b}(s',a') = 1$$

$\mathcal{C}$ **is CIS** Now suppose $\tilde{b}$ is non-trivial.

We begin by showing *(i)*. By contradiction, assume $\mathcal{C}$ is not control invariant, i.e.:

$$\forall \pi, \exists s_0 \in \mathcal{C}, \exists t \geq 0 : \mathcal{F}_t^\pi(s_0) \not\subset \mathcal{C}.$$

We consider the "safest" policy that stems from $\tilde{b}$, only taking actions such that $\tilde{b}(s,a) = 0$. More generally, we consider any policy $\tilde{\pi}$ that satisfies:

$$\tilde{\pi}(s) : \forall s \in \mathcal{S}, \quad \mathrm{Supp}\left[\tilde{\pi}(\cdot|s)\right] \subseteq \arg\min_{a \in \mathcal{A}} \tilde{b}(s,a). \tag{32}$$

We have $\mathcal{F}_t^{\tilde{\pi}}(s_0) \not\subset \mathcal{C}$ for some $t \geq 0$. Hence there is a transition $(s,a,s')$ such that $s \in \mathcal{C}, a \in \arg\min_{a' \in \mathcal{A}} \tilde{b}(s,a')$ and $s' = F(s,a) \notin \mathcal{C}$. Therefore:

$$0 \stackrel{(s \in \mathcal{C})}{=} \tilde{b}(s,a) = \max\left\{i(s), \min_{a'} \tilde{b}(s',a')\right\} \geq \min_{a'} \tilde{b}(s',a') \stackrel{(s' \notin \mathcal{C})}{=} 1,$$

which is a contradiction. Hence $\mathcal{C}$ is control invariant—and moreover, the policy defined above renders it invariant (this shows *(iii)*).

Now to show that $\mathcal{C}$ is safe, again assume by contradiction $\mathcal{C}$ is not safe, i.e.:

$$\forall \pi, \exists s_0 \in \mathcal{C}, \exists t \geq 0 : \mathcal{F}_t^\pi(s_0) \cap \mathcal{G} \neq \emptyset.$$

Consider once again a "safest" policy as defined in equation 32 (that renders $\mathcal{C}$ invariant). This policy along with the non-empty intersection in the previous equation implies that:

$$\exists s \in \mathcal{C}, \ a \in \mathrm{Supp}\left[\pi(\cdot|s)\right], \ t \geq 0 : s' = F(s,a) \in \mathcal{F}_t^\pi(s_0) \cap \mathcal{G} \implies$$

$$0 \stackrel{(s \in \mathcal{C})}{=} \tilde{b}(s,a) = \max\left\{i(s), \min_{a' \in \mathcal{A}} \tilde{b}(s',a')\right\}$$

$$\geq \min_{a' \in \mathcal{A}} \tilde{b}(s',a') = \min_{a' \in \mathcal{A}} \max\left\{i(s'), \min_{a'' \in \mathcal{A}} \tilde{b}\left(F(s',a'),a''\right)\right\}$$

$$= \max\left\{i(s'), \min_{a',a'' \in \mathcal{A}} \tilde{b}\left(F(s',a'),a''\right)\right\} \geq i(s') \stackrel{(s' \in \mathcal{G})}{=} 1,$$

which is a contradiction. Hence $\mathcal{C}$ is safe.

**Maximality of the CIS** We finish by showing *(ii)*. Assume (by contradiction) $\mathcal{C}$ is unreachable from outside. Assume furthermore $i(s) = 0$ (if $i(s) = 1$, this would mean $s \in \mathcal{G}$).
$\mathcal{C}$ reachable from outside means:

$$\exists s \notin \mathcal{C}, \exists a \in \mathcal{A} : s' \triangleq F(s,a) \in \mathcal{C} \implies$$

$$1 \stackrel{(s \notin \mathcal{C})}{=} \min_a \tilde{b}(s,a) \leq \tilde{b}(s,a) = \max\left\{i(s), \min_{a'} \tilde{b}(s',a')\right\} \stackrel{i(s)=0}{=} \min_{a'} \tilde{b}(s',a') \stackrel{(s' \in \mathcal{C})}{=} 0$$

$\square$

## A.4 PROOF OF THM. 2

*Proof:* Throughout this proof, similar to the proof of Theorem 1, we will make use of the following alternative representation of fixed points of the binary Bellman operator $\mathcal{O}$.

**Lemma 2** $\tilde{b}$ *is a fixed point of $\mathcal{O}$ if and only if it satisfies, for all $s \in \mathcal{S}$, for all $a \in \mathcal{A}$:*

$$\tilde{b}(s,a) \geq \max \left\{ i(s), \min_{a' \in \mathcal{A}} \tilde{b}(s',a') \right\} \quad \text{where } s' = F(s,a) . \tag{33}$$

*Proof:*

$$\tilde{b}(s,a) \geq i(s) + \big(1 - i(s)\big) \min_{a'} \tilde{b}(s',a') = \begin{cases} i(s), & i(s) = 1 \\ \min_{a'} \tilde{b}(s',a'), & i(s) = 0 \end{cases}$$

$$= \max \left\{ i(s), \min_{a' \in \mathcal{A}} \tilde{b}(s',a') \right\}$$

$\square$

Now, to the main proof.

**Spourious fixed point**  Firstly, note that $\tilde{b} \equiv 1$ is indeed one possible fixed point of equation 3:

$$\forall (s,a) \in \mathcal{S} \times \mathcal{A}, \quad \tilde{b}(s,a) = 1 \geq i(s) + \big(1 - i(s)\big) = i(s) + \big(1 - i(s)\big) \min_{a'} \tilde{b}(s',a')$$

**$\mathcal{C}$ is CIS**  Now suppose $\tilde{b}$ is non-spourious.

We begin by showing *(i)*. By contradiction, assume $\mathcal{C}$ is not control invariant, i.e.:

$$\forall \pi, \exists s_0 \in \mathcal{C}, \exists t \geq 0 : \mathcal{F}_t^\pi(s_0) \not\subset \mathcal{C}.$$

We consider the "safest" policy that stems from $\tilde{b}$, only taking actions such that $\tilde{b}(s,a) = 0$. More generally, we consider any policy $\tilde{\pi}$ that satisfies:

$$\tilde{\pi}(s) : \forall s \in \mathcal{S}, \quad \text{Supp}\,[\tilde{\pi}(\cdot|s)] \subseteq \arg\min_{a \in \mathcal{A}} \tilde{b}(s,a). \tag{34}$$

We have $\mathcal{F}_t^{\tilde{\pi}}(s_0) \not\subset \mathcal{C}$ for some $t \geq 0$. Hence there is a transition $(s,a,s')$ such that $s \in \mathcal{C}, a \in \arg\min_{a' \in \mathcal{A}} \tilde{b}(s,a')$ and $s' = F(s,a) \notin \mathcal{C}$. Therefore:

$$0 \stackrel{(s \in \mathcal{C})}{=} \tilde{b}(s,a) \geq \max\left\{ i(s), \min_{a'} \tilde{b}(s',a') \right\} \geq \min_{a'} \tilde{b}(s',a') \stackrel{(s' \notin \mathcal{C})}{=} 1,$$

which is a contradiction. Hence $\mathcal{C}$ is control invariant—and moreover, the policy defined above renders it invariant (this shows *(iii)*).
Now to show that $\mathcal{C}$ is safe, again assume by contradiction $\mathcal{C}$ is not safe, i.e.:

$$\forall \pi, \exists s_0 \in \mathcal{C}, \exists t \geq 0 : \mathcal{F}_t^\pi(s_0) \cap \mathcal{G} \neq \emptyset.$$

Consider once again a "safest" policy as defined in equation 34 (that renders $\mathcal{C}$ invariant). This policy along with the non-empty intersection in the previous equation implies that:

$$\exists s \in \mathcal{C},\ a \in \text{Supp}\,[\pi(\cdot|s)],\ t \geq 0 : s' = F(s,a) \in \mathcal{F}_t^\pi(s_0) \cap \mathcal{G} \implies$$

$$0 \stackrel{(s \in \mathcal{C})}{=} \tilde{b}(s,a) \geq \max\left\{ i(s), \min_{a' \in \mathcal{A}} \tilde{b}(s',a') \right\}$$

$$\geq \min_{a' \in \mathcal{A}} \tilde{b}(s',a') \geq \min_{a' \in \mathcal{A}} \max\left\{ i(s'), \min_{a'' \in \mathcal{A}} \tilde{b}\left(F(s',a'),a''\right) \right\}$$

$$= \max\left\{ i(s'), \min_{a',a'' \in \mathcal{A}} \tilde{b}\left(F(s',a'),a''\right) \right\} \geq i(s') \stackrel{(s' \in \mathcal{G})}{=} 1,$$

which is a contradiction. Hence $\mathcal{C}$ is safe. $\square$

**Remark 1** *Although not completely formal, we can argue that the set $\mathcal{C}$ associated with a fixed point $\tilde{b}$ of $\mathcal{O}$ is a subset of some set $\tilde{\mathcal{C}}$ associated with a fixed point $b$ of $\mathcal{T}$: Starting from $\mathcal{C}_0 = \mathcal{C}$, we define*

$$\mathcal{C}_k = \mathcal{C}_{k-1} \cup \{s : s \notin \mathcal{C}_{k-1}, \exists a \in \mathcal{A}, F(s,a) \in \mathcal{C}_{k-1}\}, \tag{35}$$

*and let $\tilde{\mathcal{C}} = \cup_{k=0}^{\infty} \mathcal{C}_k$. Let*

$$b(s,a) = \begin{cases} 0, & s \in \tilde{\mathcal{C}}, F(s,a) \in \tilde{\mathcal{C}} \\ 1, & o.\ w. \end{cases} \tag{36}$$

*then this $b$ function is a fixed point of $\mathcal{T}$, corresponding to a CIS set $\tilde{\mathcal{C}}$ that is a superset of $\mathcal{C}$. As a result, whenever $\mathcal{C}$ is not maximal, it is always possible to find a superset that is also a fixed point.*

INVERTED PENDULUM

Table 1: Hyperparameters for inverted pendulum experiment

|  | B2E (Ours) | SBE |
|---|---|---|
| State space dimension | 3 | |
| Action space cardinality | 5 | |
| NN hidden layers | [256, 256] | [256,256] |
| NN activation | [Tanh, Tanh, Sigmoid] | [Tanh, Tanh] |
| Learning rate[†] | $(1 - p) \times 10^{-4} + p \times 10^{-6}$ | |
| Optimizer | Adam | |
| Discount $\gamma$ | N/A | 0.9999 |
| Exploration factor | 1 | $\max\{0.95 \times 0.6^p, 0.05\}$ |
| DDQN update | N/A | Hard every 10 episodes |
| Buffer size | 50000 | |

[†]$p \triangleq$ progress, the fraction between the current episode and the total episodes.

**SBE safety critic**  For Safety Bellman equation (SBE) (Fisac et al., 2019), the MDP at each step returns the signed distance to the unsafe set $h(s) = \frac{\pi}{2} - |\theta|$. The algorithm learns $q(s, a)$, and in principle any $(s, a)$ such that $q(s, a) \geq 0$ is safe.

A more conservative safety-critic is one such that $q(s, a) \geq \eta$ for some $\eta > 0$. When evaluating the learned models (Fig. 4, right) we consider different policies $\pi_\eta$, defined as the uniform-safe over the presumed safe actions (similar to our case):

$$\pi_\eta(a|s) = \begin{cases} 0 & \text{if } q_\eta(s, a) < \eta \\ 1/\sum_{a' \in \mathcal{A}} \mathbb{1}\{q^\theta(s, a') \geq \eta\} & \text{if } q^\theta(s, a) \geq \eta \end{cases}$$

AUTONOMOUS DRIVING

The observation space is 25 dimensional and has the position $(x, y)$ and velocities in each axis, relative to the ego vehicle, plus an extra binary value representing whether that vehicle is on the screen. The action space is discrete and corresponds to five meta-actions: `slower`, `faster`, `left lane`, `right lane`, `idle`. More details at .

Table 2: Hyperparameters for inverted pendulum experiment

|  | B2E (Ours) | SBE |
|---|---|---|
| State space dimension | 25 | |
| Action space cardinality | 5 | |
| NN hidden layers | [256, 256, 256] | [256, 256, 256] |
| NN activation | [Tanh, Tanh, Tanh, Sigmoid] | [Tanh, Tanh, Tanh, Tanh] |
| Learning rate[†] | $(1 - p) \times 10^{-4} + p \times 10^{-6}$ | |
| Optimizer | Adam | |
| Discount $\gamma$ | N/A | 0.98 |
| Exploration factor | N/A | $\max\{0.95 \times 0.6^p, 0.05\}$ |
| Buffer size | 100000 | |

**Our algorithm**  We take as $\mathcal{D}_{\texttt{safe}}$ 100 initial states with the corresponding action `slower`.

**PPO**  Additional hyperparameters are the standard ones taken from (Raffin et al., 2021)

