# OpenReview forum: "Correct-by-design Safety Critics using Non-contractive Binary Bellman Operators"
_ICLR.cc/2024/Conference — ICLR 2024 Conference Withdrawn Submission_

### Official Review · Reviewer_XLgV · 2023-10-27

**Soundness:** 2 fair
**Presentation:** 2 fair
**Contribution:** 1 poor
**Rating:** 3
**Confidence:** 4

**Summary:**

The authors present a framework for obtaining correct-by-design safety critics in RL to a region of the unsafe, "failure" state space. This framework exploits the logical safe/unsafe nature of the problem and yields binary Bellman equations with multiple fixed points. These fixed points are meaningful, by characterizing their structure in terms of guaranteeing safety and maximality. The authors empirically evaluate the safety performance of their proposed method.

**Strengths:**

- Good theoretical results. However, this theoretical results are based on a specific problem settings, so I have several concerns as written in Weakness
- This paper is easy to follow.

**Weaknesses:**

### Problem setting
I am not fully convinced whether the problem setting of this paper is really useful. In this paper, there is no notion of reward and an agent receives a feedback of safety as a binary value. I do feel that typical constrained MDP formulations that are characterized by both reward and safety are much more useful. I personally would like the authors to discuss why this problem settings are considered and its advantages while comparing to:
    - CMDP with reward and safety feedback
    - MDP with binary reward feedback
    - Methods in control theory (e.g., CBF, CLF)
I know the authors partially discuss the relevance, but I do not think that the advantages of the authors' problem settings and method are well presented.

Also, the authors consider a deterministic state transition $F$ and discrete action space $\mathcal{A}$, but this assumption also limits the applicability of this paper.

In order to make this paper more useful, I would recommend the authors to reconsider the problem settings.

### Strongly related work

As a strongly related work, Turchetta et al. (2016) exists. This paper considers considers similar MDP as this paper (except for safety feedback is numeric) and defines reachability and returnability sets. I think the authors should discuss what differences and advantages exist. Also, the authors paper may need to consider returnability as well as reachability in order to guarantee the long-term safety.

- Turchetta, Matteo, Felix Berkenkamp, and Andreas Krause. "Safe exploration in finite markov decision processes with gaussian processes." Advances in neural information processing systems 29 (2016).

### Empirical experiments

I am not fully satisfied with the experiments in terms of difficulties of environments, baseline methods, and results. Regarding Inverted Pendulum tasks, if I understand correctly, the task is to stabilize the system. I do not consider that the performance of the authors' proposed method is not well presented while comparing it with SBE in this easy task. In such a well-known control task, the baseline methods should include CBF methods in control theory literature. Also, in autonomous driving tasks, I think that PPO is not a good baseline method to be compared. PPO is a safety-agnostic method and it is unfair to compare the safety performance of the PPO and the authors' proposed method.

Also, the theoretical and empirical results seem inconsistent (I do not say that they are wrong). If I understand correctly, the authors' theoretical claims regarding fixed points are core contributions, but this point is totally ignored in the empirical evaluations. I am confused what claims the authors want to support by the experiments.

Finally, the experiment section and its appendix are quite confusing.
In Appendix A.5, both Tables 1 and 2 have captions of "Hyperparameters for inverted pendulum experiment". In Table 2 (I guess this is a table for autonomous driving), SBE (not PPO) is in the algorithm list.


### Minor Comments
- In the first paragraph in Introduction, citations look weird.
    - Yu et al. (2021),Brunke et al. (2022) --> (Yu et al., 2021; Brunke et al., 2022)

- Please define $\mathbb{P}^\pi$. I know what it is, but please do not assume every reader knows that. This is also true with $E_\pi$.

**Questions:**

[Q1] Could you tell me that reason why the problem settings are considered in this paper based on my commentws?

[Q2] Why returnability is not considered? Is it possible to guarantee a long-term safety?

---

### Official Review · Reviewer_A9xS · 2023-10-30

**Soundness:** 2 fair
**Presentation:** 3 good
**Contribution:** 2 fair
**Rating:** 3
**Confidence:** 4

**Summary:**

Existing approaches to learning safety critics treat them as exactly analogous
to standard Q functions. However, this view loses sight of the high assurance
required by safety critics, especially in cases where deterministic safety
guarantees are desirable. This paper proposes a new approach to safety critic
learning based on binary Bellman equations which avoids discount factors in
order to give stronger guarantees about the resulting function. Theoretical
results show that while there is not a unique fixpoint for safety critics
defined this way, (nearly) any fixpoint provides the desired safety guarantees.
Moreover, fixpoints are "maximal" in a certain sense which allows for freer
exploration. The experiments provide evidence that for similar levels of safety,
the proposed approach is able to maintain higher policy entropy compared to
prior work.

**Strengths:**

The theoretical contribution is interesting and significant. Analyzing safety
critics using tools developed for analyzing reward signals does indeed lose some
ignore some facets of the problem. The maximality result in Theorem 1, part (ii)
is quite important.

The results for the pendulum benchmark are promising. In particular, the right
side of Figure 4, showing that policies can achieve both safe behavior and high
entropy, is impressive. This has traditionally been a challenging trade-off for
safe RL research where safety guarantees usually come with restrictions that
limit exploration.

**Weaknesses:**

The restriction to deterministic environments with finite action space is quite
limiting. Most other work in this space does not have such restrictions, though
they may be necessary for the kind of strong safety guarantees this paper is
pursuing.

The paper explores only two benchmark environments making it very difficult to
judge its empirical effectiveness. Of those, one is an extremely low-dimensional
pendulum system. The results in the more complex environment show minimal
improvement over an unsafe RL baseline (PPO).

**Questions:**

I was not quite sure where the one-sided operator appeared in the
implementation. Specifically, Algorithm 1 requires that $b^\theta$ can correctly
classify all data points as safe or unsafe, but the one-sided operator only
requires that $b^\theta$ never categorizes a safe point as unsafe (that is, it
may overestimate the danger at any point but never underestimate). Is that
behavior assumed somewhere within the accuracy check?

---

### Official Review · Reviewer_2vzB · 2023-10-31

**Soundness:** 3 good
**Presentation:** 3 good
**Contribution:** 2 fair
**Rating:** 5
**Confidence:** 4

**Summary:**

The authors consider the problem of enforcing safety in reinforcement learning (RL) algorithms. They define a (binary) Bellman equation to characterise safety for a finite MDP with stochastic policies and deterministic transitions between states and study its properties. They then proceed by proposing an algorithm to learn a neural network to compute safety. An empirical validation on two benchmark is provided.

**Strengths:**

- The paper is clear and addresses a problem of interest for the ICLR community

- The approach is sound. Furthermore, the study of the fixed points of the proposed Bellman operator and the resulting Algorithm 1 are interesting and show promising results

**Weaknesses:**

- Bellman equations to characterise safety and equivalently reachability properties have been already developed in the literature, and also in the stochastic setting [1,2]. The one proposed by the authors is an extension of those ones, with the added constraint that only actions that have probability 1 of being safe are admitted. This should be clarified and discussed in the paper.

- The approach proposed by the authors is also related to barrier functions [3,4] and in particular neural barrier functions [5], where these functions are employed to compute regions of the state space that are safe (for a finite or infinite horizon). Pros and Cons of the approach of the authors wrt barrier functions should be discussed in the related works

- The authors claim that the fact that the operator in Eqn 4 has multiple fixed points is surprising. This is actually to be expected. In fact, it is well known that fixed points for safety properties will in general depend on initial condition and topology of the graph of the MDP [6].

- I find Figure 5 and its analysis a bit unclear. In fact, the authors stop the simulations at 700 k iterations at a point where the proposed approach was slightly outperforming  PPO. However, in the previous iterations, PPO was almost always giving better performances. So, can you please run the experiment for more iterations? Say at least 1 milion.

[1] Abate, Alessandro, et al. "Probabilistic reachability and safety for controlled discrete time stochastic hybrid systems." Automatica 44.11 (2008): 2724-2734.

[2] Laurenti, Luca, et al. "Unifying Safety Approaches for Stochastic Systems: From Barrier Functions to Uncertain Abstractions via Dynamic Programming" arXiv preprint arXiv:2310.01802, 2023

[3] Zeng, Jun, Bike Zhang, and Koushil Sreenath. "Safety-critical model predictive control with discrete-time control barrier function." 2021 American Control Conference (ACC). IEEE, 2021.

[4] Prajna, Stephen, Ali Jadbabaie, and George J. Pappas. "A framework for worst-case and stochastic safety verification using barrier certificates." IEEE Transactions on Automatic Control 52.8 (2007): 1415-1428.

[5] Mathiesen, Frederik Baymler, et al. "Safety certification for stochastic systems via neural barrier functions." IEEE Control Systems Letters 7 (2022): 973-978.

[6] Baier, Christel, and Joost-Pieter Katoen. Principles of model checking. MIT press, 2008.

**Questions:**

Please, see Weaknesses Section

---

### Official Review · Reviewer_WKp6 · 2023-10-31

**Soundness:** 3 good
**Presentation:** 3 good
**Contribution:** 2 fair
**Rating:** 5
**Confidence:** 3

**Summary:**

This paper studies how to use Bellman-like operators to learn safe policies and critics to avoid reaching unsafe states. Unlike the standard RL setting, where the performance of policies can be quantified with rewards, the safety problem that this paper is concerned with only evaluates the safety of a policy in a binary manner. This paper formulates the Bellman operators with real value operations to achieve the binary operations on the binary outcomes. Furthermore, to overcome the non-contraction problem of the proposed Bellman operator, the paper proposes to learn a one-sided Bellman operation of which the fixed points are more conservative in staying in the safe region. The paper developed an algorithm by learning the policy that corresponds to the fixed-point of this one-sided problem iteratively.
The paper demonstrates the effectiveness of the proposed method on two benchmarks.

**Strengths:**

* `originality`: This paper proposes a Bellman operators for binary outcomes that is unlike the conventional Bellman operations for RL.
* `quality`: The paper is well written.
* `clarity`: The problem definitions are clear. The Bellman operator proposed in this paper is sound.
* `significance`: This paper provides a different choice than the conventional ones for safe RL.

**Weaknesses:**

* `Weakness 1`: It is questionable whether the proposed approach is still valid in a stochastic environment, where there is always a chance of entering an unsafe region regardless of the state that the agent is in. While in a simulated environment, the dynamics can be deterministic, it is essential to acknowledge that stochasticity is prevalent in the real world.
* `Weakness 2`: It is also questionable whether the proposed approach is valid when the goal does not perfectly align with safety. In the autonomous driving benchmark, the goal is to avoid collision. But in reality, not every task is about set avoidance or reachability.
* `Weakness 3`: in the experimental section, the author uses only two benchmarks to validate the proposed approach. It would be better if the author demonstrated how the proposed approach can be adopted to solve different tasks. Also, the baseline RL approach hinges on the reward function. Different reward functions can lead to different performances. It would be better if the author just tried to use a goal-driven reward (+1 reward for reaching the goal -1 for collision, and 0 everywhere else) to train the RL policy for comparison.

**Questions:**

Please address my questions in `Weakness 1` and `2` in the `Weakness` field.

---

### Author Response · Authors · 2023-11-22
**Response & reason for withdrawal**

We thank the reviewers for their constructive criticism. They have highlighted that characterizing safety from the perspective of the fixed points of the Binary Bellman Operators is a theoretical strength, and that our experiments show promising results in terms of being able to learn safe policies with high entropy. In spite of this, they raised several concerns and offered suggestions for improvement. In consideration of our limited time, we have decided to withdraw the submission and focus our efforts on addressing these concerns and improving our work.

Their main unanimous concerns are two-fold. Firstly, assuming deterministic transitions is a drawback and may be seen as unrealistic. We agree, and are confident that the setting can be relaxed to stochastic systems with bounded disturbances. However (as one reviewer pointed out) in the presence of unbounded noise we are likely to lose our guarantees, which are inherently posed in a worst-case sense. Secondly, they perceive the experimental validation was insufficient. We also agree that the comparison against PPO is unfair, and that we should compare with more safety critics in the literature. On a final note, in light of the reviews, it is clear we failed to convey the notion that our safety critic can be learned in parallel with any other reward optimization algorithm, and so is applicable to any MDP with reward signals as well.

Since running comprehensive experiments and extending our theory to the stochastic setting would necessitate more time than what is allocated for this revision, we have decided to withdraw the paper. We thank the reviewers again for their valuable feedback.